# Generation of 3D Cardiovascular Ultrasound Labeled Data via Deep Learning

**Cristiana Tiago**[1,2]                                           CRISTIANA.TIAGO@GE.COM
**Andrew Gilbert**[1,2]                                            ANDREW.GILBERT@GE.COM
**Sten Roar Snare**[1]                                            STENROAR.SNARE@GE.COM
**Jurica Sprem**[1]                                               JURICA.SPREM@GE.COM
**Kristin McLeod**[1]                                             KRISTIN.MCLEOD@GE.COM

[1] *GE Vingmed Ultrasound, GE Healthcare, Horten, Norway*

[2] *Department of Informatics at the University of Oslo, Oslo, Norway*

**Editors:** Under Review for MIDL 2021

## Abstract

We propose an image generation pipeline to synthesize 3D echocardiographic images with corresponding ground-truth labels, to alleviate the need for data and for laborious and error-prone human labeling of images for subsequent Deep Learning (DL) tasks. The proposed method relies on detailed anatomical models, obtained from CT, of the heart as ground-truth label sources. These models are used to extract labeled slices which, together with a second dataset made of real 3D echocardiographic images, allow to train a Generative Adversarial Network (GAN) – namely, a CycleGAN –to synthesize realistic 3D cardiovascular ultrasound images that are paired with ground-truth labels. A qualitative analysis of the synthesized images showed that the main structures of the heart are well delineated and closely follow the labels from the anatomical models, making it possible to use these 3D echocardiographic images and paired labels for training new DL 3D segmentation or landmark detection models.

**Keywords:** 3D image generation, cardiovascular ultrasound, CycleGAN, deep learning, echocardiography, generative adversarial networks

## 1. Introduction and State of the art

Compared to 2D, 3D echocardiography images are more complicated to acquire and can be used for more accurately measuring ventricles and atria volumes and performing 3D stress imaging. Obtaining these images with the required quality to perform such measurements is critical to ensure accurate results. GANs are a type of artificial neural networks (NNs) that can generate new images matching prior data distributions. The GANs are built from a generator and a discriminator, where the former is responsible for generating new images, and the latter distinguishes between the generated images and real ones to give feedback during training. Besides generating new data similar to a given input dataset, GANs can perform domain translation, i.e. given an instance from one training domain, the network can synthesize an equivalent instance belonging to a second domain and vice-versa. The main strength of CycleGAN (Zhu et al., 2017) is that a dataset of paired images is not required to perform domain translation.

(Abbasi-Sureshjani et al., 2020) developed a method to generate 3D labeled Cardiac Magnetic Resonance images relying on anatomical models to obtain labels for the synthesized images. Similarly, (Gilbert et al., 2021) proposed an approach to synthesize labeled

2D echocardiography images, using anatomical models and a CycleGAN. Both groups investigated the use of generated datasets to train NNs to segment different cardiac structures on different imaging modalities.

The main contribution brought up by this work is the 3D extension of (Gilbert et al., 2021) work to include a pipeline to generate labeled 3D echocardiographic datasets, using anatomical models from CT and a CycleGAN.

## 2. Methodology and Results

This work focuses on the data generation pipeline extension to 3D. As demonstrated by (Amirrajab et al., 2020), a GAN can be trained on 2D slices and be used on 3D cases at inference time. To generate the image data, a CycleGAN was trained on 2 unpaired image domains: pseudo cardiovascular ultrasound and real images. The former image domain was made of pseudo images obtained from 3D anatomical heart models, as described in (Gilbert et al., 2021). The heart models considered one time point during the cardiac cycle, end diastole in this work, and were sliced consecutively by translating the plane initially aligned with the long axis of the heart and 820 images resembling ultrasound, i.e. pseudo, were created, corresponding to 20 3D images (41 2D slices per 3D model). The later image domain consisted of 16 sliced 3D images, acquired with GE Vivid Ultrasound Systems, creating a dataset with 656 slices. At inference time, the CycleGAN generated 41 consecutive 2D slices, corresponding to a single 3D image. The CycleGAN training on a NVIDIA GeForce RTX 2080 Ti graphics card took around 72h and at inference time each 3D image (41 slices) was generated in 3s.

Figure 1 shows the methodology used in this work, from the anatomical models preprocessing to the inference results obtained. A qualitative analysis of the synthesized images indicates that the main structures of the heart are well delineated in the generated images. Moreover, image details such as the cone, noise and speckle patterns are also present. 4 consecutive slices of 2 generated 3D images at inference time together with the paired labels are also shown. The delineation of the left ventricle is shown for illustrative purposes, however any cardiac structure considered in the anatomical models could be annotated similarly. The pipeline synthesizes 3D echocardiographic datasets with corresponding labels delineating different structures.

## 3. Conclusion

A pipeline to generate labeled 3D echocardiography images was proposed. Qualitative analysis of the images suggests that the results obtained from the CycleGAN are realistic and consistent when generating consecutive slices of a volume. The results presented indicate that the proposed approach has potential to tackle the lack of availability of labeled medical data since it can be used, at inference time, to generate labeled training samples from unlabeled 3D cardiovascular ultrasound datasets. Further experiments are required to extend this work to the whole cardiac cycle, generating 4D (3D+time) images.

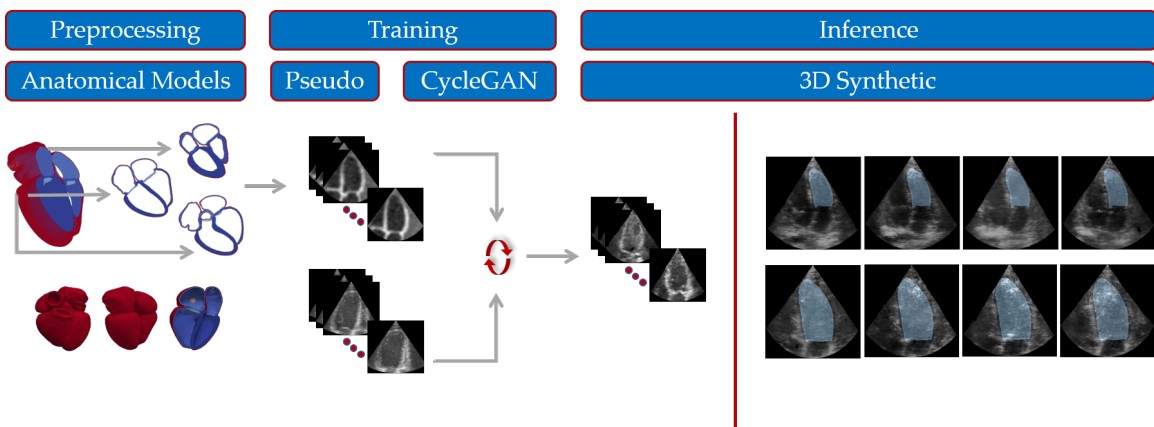

Figure 1: 3D echocardiography image generation pipeline and inference results. During preprocessing the anatomical models are sliced (41 slices) consecutively and 2D pseudo images are created from these slices. To train the CycleGAN, the pseudo images are used together with real 3D images, also sliced. At inference time, the CycleGAN generates 41 slices corresponding to one 3D image. On the right side of the figure, 2 examples obtained at inference time are shown. For each 3D synthetic image, 4 consecutive slices of these volumes are shown together with a left ventricle label. The proposed method is able to generate physiologically realistic images, giving correct structural features and image details.

## Acknowledgments

This project received funding from the European Union's Horizon 2020 research and innovation programme under the Marie – Skłodowska – Curie grant agreement No 860745.

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
