# OpenReview forum: "Generation of 3D Cardiovascular Ultrasound Labeled Data via Deep Learning"
_MIDL.io/2021/Conference/Short — Submitted to MIDL 2021_

### Official Review · Reviewer_9KhX · 2021-04-28

**Confidence:** 4
**Final Rating:** 2

**Summary:**

The authors propose a method to simulate realistic 3D echo data from CT images. The paper is well written and clear, and the methodology is appropriate for the task. My main concern is about the novelty: the method seems like a straightforward extension of a 2D framework by the same group. The reason why I think this may be an issue is that echo -specific differences between 2D echo and 3D echo (such as difference in resolution, difference in the density of spatial sampling and fan-like elevation sweeping to create a 3D volume to cite the main ones) have been ignored, while they should have been what makes this contribution novel. Maybe this actually has been taken into account by authors and has been omitted from the text for length limits, in which case authors need to clarify this.

**Strengths:**

* The results shown in the paper look highly realistic
* The proposed method would allow creation of a very large database of annotated 3D echo data,  which would be a very useful resource for developing data-driven methods such as segmentation, classification and derivation of biomarkers form echo images.

**Weaknesses:**

* Novelty seems to be limited to a 3D extension of previous 2D work, and this seems to have been done by just replacing 2D layers to 3D layers

* Evaluation is weak: quantitative analysis of the results is limited to the execution time, and qualitative results is limited to a rather small figure showing 8 examples of central slices of 3D volumes.

**Deanonymize Review:**

yes

**Detailed Comments:**

Authors first describe the clinical need (and challenge) of collecting high quality 3D echo data. However, the proposed work is on how to generate realistic synthetic 3D echo data. Then there is no connection on link on how can realistic simulations help acquire better 3D volumes. I have the impression that they want to convey that the proposed simulations can help train better segmentation algorithms that will work even if the data is not that good in the first place? Although I see how any method to improve segmentations and possibly give access to more labeled data could be beneficial, this needs to be very clearly stated in the introduction as currently the link between the application and the proposed work is not obvious.

Given the limited length of the paper, I would recommend removing the description of what a GAN does (which is common knowledge for the MIDL community) and make use of the space to better describe what the proposed method is beneficial for.

It is not clear from the paper if the slices that make up the 3D image are computed as a stack (e.g. perpendicular to the y direction) or as a fan i.e, in spherical coordinates following the elevation angle direction). This may not have a visual impact on the central slice (the only one shown in the examples) but I expect it to have a high impact in all other slices and particularly on the ability of the network to reproduce view-direction artefacts such as shadows, because if stack-like slicing is used then information about artefact-generating tissues near the transducer may be missing.

It would be nice to see a more thorough evaluation, for example: can experts differentiate between real and synthetic images? What is the accuracy image segmentation methods when trained on synthetic images only?

**Justification Of The Rating:**

I think this work is a straightforward extension of 2D work previously proposed by the same group, so the technical novelty is very limited. This could be compensated by application novelty, or solid evaluation, unfortunately the application is not particularly clear and the evaluation is weak.

**Paper Type:**

methodological development

**Special Issue:**

no

---

### Official Review · Reviewer_sTmk · 2021-04-29

**Confidence:** 4
**Final Rating:** 1

**Summary:**

This paper proposes to circumvent the difficulty of collecting annotated 3D echocardiographic images (here annotations mean segmented end-diastolic frames, for segmentation purposes), by the use of a CycleGAN, which is relevant to generate realistic images without the need for paired data. They generate a stack of 2D slices that correspond to a single 3D frame.

**Strengths:**

- A timely framework to generate realistic images and therefore avoid the cumbersome annotations.
- The use of CycleGANs, which are very relevant for the purpose of this application.
- Addressing 3D echocardiographic data.

**Weaknesses:**

- The focus on end-diastolic frames only, and the generation of 2D slices (meaning that claims on 3D may be softened).
- Limited/no methodological contribution, not compensated by a thorough evaluation (here, very qualitative comments on 2 examples).
- A strategy that is valuable for segmentation but limited to this specific purpose.

**Deanonymize Review:**

no

**Detailed Comments:**

To complement my remarks above:
- The title should mention the segmentation purpose, to which this work is specific.
- Although the paper length is limited, the literature review could briefly comment on complementary strategies to synthetize realistic images (computational models instead of anatomical models, and registration or physics-based image simulation instead of GANs).
- The fact that 3D images are not directly reconstructed (but that 2D stacks are used) should be acknowledged and discussed.
- It is not fully clear to me how the variety of models was obtained: do they correspond to different subjects? Could the authors better describe this?
- Even for a short paper, the evaluation could be less qualitative. What about estimating the segmentation error on these synthetic data, or the improvement in segmenting real images by using these synthetic images?

Writing:
- Abstract: “to alleviate the need for data” could specify “annotated data”

**Justification Of The Rating:**

Although the purpose is timely and the authors use relevant methods, the methodological contribution is limited, and not supported by a clear demonstration in full 3D and/or a quantitative evaluation.

**Paper Type:**

validation/application paper

**Special Issue:**

no

---

### Meta-Review · Program_Chairs · 2021-05-06

**Recommendation:** Reject
**Confidence:** 5

**Metareview:**

Both reviewers give reject, with major concern on limited methodology novelty.

---

### Decision · Program_Chairs · 2021-05-11

Reject